# Engagement of Preschool-Aged Children in Daily Routines

**DOI:** 10.3390/ijerph192214741

**Published:** 2022-11-09

**Authors:** Špela Golubović, Mirjana Đorđević, Snežana Ilić, Željka Nikolašević

**Affiliations:** 1Department of Special Education and Rehabilitation, Faculty of Medicine, University of Novi Sad, 21000 Novi Sad, Serbia; 2Faculty of Special Education and Rehabilitation, University of Belgrade, 11000 Belgrade, Serbia; 3Department of Psychology, Faculty of Medicine, University of Novi Sad, 21000 Novi Sad, Serbia

**Keywords:** engagement, children with developmental disabilities, preschool-aged

## Abstract

Child engagement refers to the time spent interacting with physical and social environments according to age, abilities, and a situation. The aim of this study is to assess the functioning of children in early childhood routines using engagement assessment instruments relative to the presence of developmental disabilities, age, gender, and parental characteristics within the contexts of preschool and family routines. The sample comprised 150 children aged 3–5 (AS = 4.02, D = 0.78), including typically developing children (N = 49) and children with developmental disabilities (N = 101). To assess the children’s engagement in preschool classrooms, we used the Classroom Measure of Engagement, Independence, and Social Relationships (ClaMEISR), and the Child Engagement in Daily Life Measure was used to assess the children’s engagement in family routines. The results obtained indicate a significantly higher rate of engagement in routines and activities among girls and older children. Parental characteristics associated with children’s engagement included employment and marital status. Children with developmental disabilities, compared to their typically developing peers, had lower levels of engagement in social relationships and functional independence in daily routines. The results indicate that both instruments have a high internal consistency and are thus suitable for future use in the Republic of Serbia.

## 1. Introduction

Fully inclusive and qualitative programs in early childhood education should meet children’s needs and provide all children with opportunities to actively participate in classroom activities and routines [1]. The two subdomains of participation, involvement and engagement, are regarded as key priority outcomes of inclusive practices in early childhood education. As stated in the literature, involvement refers to an internal state of interest towards an activity itself, while engagement refers to the specific behavior, emotions, and thoughts [2]. With regard to early childhood education, engagement is expressed through the time the child spends engaged in activities, as well as the level of interaction with the social and physical environments and timeframe in which they are achieved in a developmentally and contextually appropriate manner. Thus, children’s engagement during early childhood education mainly refers to the manipulation of objects and materials, involvement in games with peers or adults, and participation in the activities of everyday life [3,4]. On the subject of children’s engagement in daily life activities, Chiarello et al. [5] highlighted two important areas: participation in family and recreational activities and self-care, indicating that engagement in these areas provides a great deal of opportunities for having fun and communicating and interacting with other persons. In addition to engagement, independence and social relationship are the three pillars of learning that are necessary prerequisites for learning to occur [4,6]. Whether at home or in the preschool group setting, the amount of time that a child can be engaged in all their daily routines is diverse, depending on varied levels of sophistication and types (adults, peers, and materials) [7].

Through participating in diverse preschool classroom activities, a child learns and practices behaviors to promote independence, security, and manual dexterity in different daily routines and to develop supportive and enhancing relationships with the environment, within the peer group, and with the adults who contribute to the improvement of the child’s overall well-being and executive function skills. In accordance with the previously stated remarks about children’s participation in family daily routines, Savahl et al. [8] found that children’s engagement with the family and participation in daily activities explain 31% of the variance in children’s subjective well-being. The levels of a child’s engagement in daily routines and activities are related to individual child characteristics, the family environment, and environmental factors and can be a predictor of his/her further development, functioning, and school success [7,9]. A child’s individual characteristics related to his/her engagement include the child’s chronological age, disability status, and temperament, whereas classroom characteristics are related to the teacher’s functions and the structural features of the learning environment. Regarding the predictors of sophisticated engagement, the authors highlighted the child’s characteristics and, with respect to non-engagement, the predictors are associated with the environmental characteristics [10]. Children displaying low levels of engagement are at risk of later difficulties in learning, behavior, and social interaction with others [7,11].

Although emphasis has been placed on numerous positive outcomes of high-quality inclusive programs, practical experience in research shows that children with developmental disabilities have limited opportunities to engage in activities [12,13,14,15]. Peer group inclusion presents a great challenge to children with developmental disabilities, since it is related to the development of self-regulation, which basically means the ability to direct and sustain short-term attention and self-regulate emotions and behavior in response to changing environmental demands [16]. Children with poor self-regulation skills and profound levels of developmental disabilities are more often in a situation where they are excluded from joint activities with their peers, which reduces opportunities for promoting socio-emotional development, while children with speech and language delays are less involved in rule games and attend reading sessions less often in comparison to children without developmental disabilities [15]. Research by McWilliam and Bailey [17] found that children with developmental disabilities, compared with their typically developing peers, are less engaged and in lower levels. Additionally, children with developmental disabilities were found to spend less time in interactive engagement activities with educators, be less engaged with peers, be less engaged when using materials in a developmentally appropriate manner compared to children without developmental disabilities and spend more time passively nonengaged. The results of the earlier studies show that lowered levels of engagement, in combination with hyperactivity, have short-term and long-term outcomes affecting the child’s functioning and well-being [9,18]. In addition, they found that children with developmental disabilities spend less time engaging in complex tasks and that they are more often engaged in activities requiring less complex behavior [19].

Additionally, the engagement of children with developmental disabilities in daily activities was at the lowest level due to circumstances related to variations in children’s classroom engagement throughout the time of transitions. Furthermore, it was found that children were more engaged with activities during free-choice time and peer interactions than they were during teacher-structured activities [20]. Difficulties in child functioning in terms of engagement manifest as decreased participation in academic or non-academic tasks and thus require additional support. Therefore, some authors [21,22] believe that it is necessary to design opportunities and activities that will promote and encourage the engagement of children with developmental disabilities in routines and daily activities and, therefore, children should simultaneously receive the right amount of support in order to increase their engagement.

### Local Context

In the Republic of Serbia, preschool education and upbringing are an integral part of the education system, designed for children from the ages of 6 months to the start of primary school (at seven years old). According to the data of the Statistical Office of the Republic of Serbia [23], in the school year 2021/22, out of 223,559 children who attended preschool education and training, 24.1% were children aged from six months to three years, and 75.9% were children aged from three to seven years. The preschool education program was implemented in 463 preschool institutions (163 state and 300 private). The Law on Preschool Education of the Republic of Serbia does not provide special institutions for children with developmental disabilities. However, children with developmental disabilities may exercise their right to a preschool upbringing and education with all other children according to an educational (inclusive)-group-based individual education plan or an individualized education plan, or in separate groups within preschool institutions attended only by children with disabilities affecting their development, which are called development groups. Daily social interactions and activities related to other educational groups are planned and implemented at the same level for a child who is enrolled in a development group. Previous analyses show that only 50% of children in Serbia between the ages of three and five are included in the Early Childhood Education and Care program, with only 4–10% of children from vulnerable groups. Although it is estimated that 5% of children have developmental disabilities or a disability, only 1.2% of the enrolled children have developmental disabilities or disabilities [24].

Previous practices in Serbia did not imply the use of any instrument that would enable the functional assessment of classroom activities among pre-school-aged children as an authentic way of determining the type of support that the child needs and to incorporate it, as such, into the child and family support program. Data collection for functional assessment in early childhood can be realized through the observation of the child (in regard to the child’s current functioning in activity contexts) and/or through a caregiver interview (via family input on the child’s functioning).

Following the example of earlier research, with the aim of overcoming the lack of existing instruments for measuring children’s engagement, independence, and social relationships in their routines, a functional assessment instrument was used for the first time in the present study as a contribution to the research and monitoring of children in their early years [25,26].

In this study, the aim was to use the instruments for functional assessment to measure the functioning of children’s engagement in routines relative to the presence of developmental disabilities, their age and gender, and parental and environmental characteristics. A further goal was to measure the effectiveness of the authentic instruments for assessing children’s engagement in the contexts of early childhood education and family.

## 2. Materials and Methods

### 2.1. Sample

The sample consisted of 150 preschool children aged from three to five (AS = 4.02, SD = 0.78) from various state preschool institutions located in urban areas in the northern part of Serbia. The respondents were divided into two groups: typically developing children and children with developmental disabilities. The sample of typically developing children consisted of children without clear evidence of motor, sensory, or mental disabilities, as well as children without a specifically developed individualized education plan. In the group of typically developing children, the average age was 4.12 (SD = 0.88). The children’s distribution according to gender was 62 (61%) boys and 39 (39%) girls. The group of children with developmental disabilities included children eligible for special education services and consisted of children who were mostly in development groups, which means that they were among groups composed of children with developmental disabilities (57%), whereas a smaller number of children were in inclusive groups (43%). The average age was 4.35 (SD = 1.01) in the group of children with developmental disabilities. This group of children consisted of 30 (62%) boys and 19 (38%) girls. Therefore, there were no statistically significant differences between the two groups in terms of their age and gender. Based on the data obtained from the available institutional documents related to the evaluation of, and eligibility determination for, special education, this group comprised 28 children with a severe form of autism and two children with a mild form, assessed by the Gilliam Autism Rating Scale, Third Edition, GARS-3 [27]. Another four children had been diagnosed with autism by a competent doctor, but there was no available information related to the ASD severity and form of these children. The remaining 15 children had mild intellectual disabilities according to data based on the psychologist’s documentation at the institution. In this research, the results were not interpreted in relation to disability type but rather to the group with developmental disabilities.

The language distribution of the entire sample by mother tongue was Serbian. All the children in our sample attended childcare, which means that the amount of time they spent in a preschool environment was six to eight hours.

More detailed information on the structure of the sample, regarding the sociodemographic characteristics, is presented in Table 1.

In the present research, data were collected from 109 educators working in early childhood education. The average age of the educators with higher levels of education was AS = 43.67, SD = 7.72. The educators’ working experience was AS = 7.39, SD = 5.42. There were 100 (92%) educators who had previous experience in working with children with disabilities, whereas there were 9 (8%) who did not have previous work experience.

### 2.2. Procedures

Permission from the principals of the preschool institutions was obtained before the research procedures started, as was the approval of the Ethics Committee of the Faculty of Medicine (approval no. 01-39/147/1). Next, the principals of the institutions were provided with a detailed description and the purpose of our research, as well as the instruments that would be applied. Accordingly, a personal interview was conducted with each of the principals. Then, meetings were organized for the parents, at which they were informed verbally, as well as in writing. In addition, they were provided with the informed consent form to be signed if they agreed to participate in the study, and they either filled out that form at the time or handed it in the next day to the person authorized to collect the forms at the preschool. Considering that the research involved young children who did not have full insight into its structure and the contents, the parents responsible for the children’s care signed the informed consent on behalf of their children. The anonymity of the research data was ensured.

Conversations were conducted and information was shared with educators, who were also informed about the purpose and method of the research and also signed their consent to participate.

### 2.3. Instruments

The parents and educators responded to a brief demographic questionnaire attached to the consent form. The questionnaire included data on the child’s date of birth and gender, the family size and place of residence, and the parents’ level of education and employment status, as well as data on the children’s home languages. The questionnaire designed to gather data on the educators included gender, age, length of service, experience in working with children with developmental disabilities, and level of education.

The Classroom Measure of Engagement Independence and Social Relationships (ClaMEISR) [28] was used to assess engagement in everyday routines in the preschool classroom. This scale measures three domains of the functional outcomes of engagement, independence, and social relationships in children aged from three to five years. It comprises 215 items divided into 13 subscales: arrival, music, bathroom, outside time, hand-washing, teacher-directed activities (in a circle), directed outside activities (in a circle), meal/snack time, teacher-led small group activities, tooth-brushing, story time, nap time, free play, and departure. Educators indicate their level of agreement with a given statement on a three-point Likert-type scale, where 1 indicates that the child still does not exhibit the specified behavior, 2 indicates that the behavior is sometimes exhibited, and 3 indicates that the behavior is often or always exhibited. Within each domain, the items that are rated as 3 are calculated. Different subscales have different numbers of items within the functional domains of engagement, independence, and social relationships. If a routine has <3 items in a domain, the score of that routine is not calculated (for example, handwashing has only one engagement item and one social relationship item, so that no scores are calculated for those two domains in handwashing). The informants of this instrument were the educators. With prior permission from the author (Robin McWilliam), the scale was translated into Serbian with the minor corrections required for the existing item’s wording. The scales were not cross-culturally adapted into Serbian.

The child’s engagement in everyday activities and routines was assessed using the Child Engagement in Daily Life Measure [29]. In the first section, which included 11 items, the frequency of participation in family and recreational activities was scored on a five-point Likert scale ranging from 1 (never) to 5 (very often). The same items were used for evaluating the child’s enjoyment of participation using a five-point scale ranging from 1 (not at all) to 5 (very much). The second section (7 items) referred to participation in self-care, i.e., ability to function independently in the activities of daily living (eating, dressing, using the toilets, and more), and this assessment was also carried out using a five-point scale ranging from 1 (the child does not perform the activity) to 5 (independent, another person is not required for the activity). During the data processing, the raw scores for the participation in family and recreational activities subscale and the self-care subscale were transformed into scaled scores. To determine the participation and levels of enjoyment of family and recreational activities, a raw score was used as a global measure of activity enjoyment. The informants of this instrument were parents. 

The present study showed the good psychometric properties of these instruments, which will be discussed further.

### 2.4. Data Analysis

The statistical analysis was conducted using SPSS (IBM Corp, released 2010, IBM SPSS Statistics for Windows, Version 22.0 Armonk, NY, USA: IBM Corp.). The descriptive statistics included the mean values and standard deviations, with minimum and maximum values. The normal distribution of all the measures was evaluated by skewness and kurtosis. The internal consistency or coherence of the scales was assessed using Cronbach’s alpha and mean inter-item correlation (MIIC). For the exploration of differences in the ClaMEISR and CEDL subscales between groups, Student’s independent t-test was used. Cohen’s d value was used to indicate the effect size. A commonly used method of interpretation is to refer to the effect sizes as small (d = 0.20), medium (d = 0.50), and large (d = 0.80) based on the benchmarks suggested by Cohen (1992). The strength of the linear relationships between variables was tested using Pearson’s correlations. We also performed a partial correlation analysis to investigate the associations between age and the instrument measurements while controlling for the existence of disabilities. A significance level of 5% was adopted for all the analyses.

## 3. Results

### Descriptive Statistics of the Measures Used

The descriptive and alpha reliabilities are shown in Table 2. All the measures were normally distributed with respect to skewness and kurtosis, within the range of ±2.0 (see Gravetter and Wallnau, 2014). Cronbach’s alpha for the ClaMEISR subscale produced coefficients in the range of 0.930 to 0.958. This indicated high inter-item consistency. The MIIC for these subscales showed very high values above 0.70, indicating the potential redundancy of the used items. In the case of the CEDL subscales, the MIICs were good and ranged between 0.880 and 0.899, and these subscales also showed good homogeneity with the average inter-item correlations of >0.250.

Correlations between the subscales of the ClaMEISR showed a pattern of significant association, whose intensity was high (Table 3). The correlation between the subscales of the CEDL had a moderate to high intensity. In regard to the correlations between the subscales of the two instruments, they were high and in a positive direction. The correlation coefficients between the criterion and the predictor mainly showed a negative direction of the relationship, with the intensity ranging from low to high. Gender and age achieved a statistically significant association according to all three subscales of the ClaMEISR questionnaire and participation in self-care subscale of the CEDL questionnaire. The obtained correlations were of a low to moderate intensity and in the positive direction. Therefore, girls and older children achieved higher scores on the mentioned scales, which indicates their better functionality.

The results revealed a statistically significant difference in the ClaMEISR and CDLS subscale depending on whether a child had developmental disabilities or was without disabilities. Children without disabilities had higher scores in all the domains, which indicates their better functionality (Table 4 and Figure 1, Figure 2 and Figure 3).

No differences in the level of education were found among the group of parents, whereas differences in marital status and employment were found.

## 4. Discussion

The aim of this study was to determine the effectiveness of the use of the authentic assessment instruments and evaluate the level of children’s engagement, independence, and social relationships within the context of family and preschool classroom everyday activities and routines relative to the child’s individual characteristics, such as gender, age, the presence of disabilities, and parental characteristics.

The obtained results showed that both the instruments applied have high internal consistency. In previous research, the obtained data on the internal consistency of the Child Engagement in Daily Life Measure were similar [29], while no data were found for ClaMEISR during the earlier testing of the psychometric characteristics of this instrument. The results of this study support the data on the internal consistency of the 3M Preschool Routines Functioning Scale, designed in Spain [26]. However, this is the first study of its kind that was carried out based on the possibilities of applying this instrument in this region. The high internal consistency values provided the basis for the further interpretation of the results obtained from these scales.

The parents’ role in enhancing their children’s development is crucial from birth, with the exception of priorities regarding inclusion in the daily routines of early childhood education, which vary according to child’s age and stage of development. It has been emphasized that their vital priorities include social communication, social interactions, academic skills, self-care, and community and social life, and therefore, children are more likely to be engaged in these activities [30,31]. Parents’ priorities for children of a young age focus on practicing self-care, and on the other hand, their focus for older children is on the role of social interactions with peers [5].

In this research, the amount of time children spent in everyday activities differed significantly in relation to gender differences, i.e., girls were significantly more involved in activities as compared to boys, as well as the child’s age, which is in accordance with the previously stated findings in the literature [29,32]. In the present study, differences in age manifested in activities at home, as well as in preschool. It is to be expected that younger children participate less often in different activities and that they need help to participate in self-care. The obtained differences show that, sometimes, a certain number of younger children need help during particular activities in order to complete them successfully, which can be explained by the fact that, at this age, these basic skills are yet to be mastered. Among the typically developing children, all the activity variables were in the range of the maximum values; thus, in a practical sense, the obtained differences do not significantly differentiate the younger from the older children. However, it should be emphasized that with age and the child’s development, the amount and quality of engagement in activities also increased. This can be considered as particularly significant, since some authors have identified a correlation between the child’s engagement and prosocial behavior, as well as improved learning outcomes [33].

The literature data indicate that the level of child involvement is influenced not only by the child’s characteristics but also by the family [34], and that the levels of engagement of children in everyday activities at home and in preschool differ significantly in relation to the sociodemographic characteristics of the parents (education level, marital status, employment status, number of children, family type). In this study, the same factors were analyzed, and the obtained data show that the employment status correlates with the child’s engagement in various activities in the preschool classroom, as well as at home, except for participation in self-care. Additionally, the parents’ marital status correlates with the child’s engagement in various activities in the preschool classroom but not those at home. These results are not in accordance with the results of Morales Murilo’s research [26], in which it was determined that children of single-parent families exhibit a higher level of involvement and engagement compared to children from two-parent families. Observed through the lens of employment, which can also be seen as a factor of socio-economic status, the data obtained in our sample are in accordance with the interpretation of the authors who suggested that functional skills in the area of self-care and social functioning are statistically significantly more developed in children living in families of a higher socio-economic status [35].

The results of this research confirm that children with developmental disabilities show a lower level of engagement and, therefore, participate less in family and recreational activities and self-care, but they enjoy them just as much. Children of the typical population are better adapted to activities at home and independently and actively participate in almost all these activities. This result can be explained by the additional needs and limitations that children with developmental disabilities have compared to typically developing children. The largest number of typically developed children are involved in family outdoor activities, such as shopping, going to the library, visiting relatives and friends, and playing with adults and other children, and they have a high degree of independence in terms of basic personal activities such as eating, dressing, and using toilets. Similar results based on a cohort of typically developing children were obtained by Chiarello et al. [29] in their study. Children with developmental disabilities are more often engaged in activities that involve the presence of adults, such as playing with them, going shopping or to a party or the zoo, or activities that involve sitting, as well as structured activities, such as drawing or coloring. These children have a far lower level of engagement in activities related to playing with their peers, whether indoors or outdoors. The child’s level of engagement in self-care is significantly higher when children are more functional. That is, their functional ability level is higher, which was confirmed by results of our study, in which this domain of child engagement in daily life activities at home was lower in children with developmental disabilities compared to typically developing children. The parents of children with developmental disabilities more often provide support to children in situations related to bathing, using the toilet, and dressing. Children with developmental disabilities acquire these skills at a slower pace, and they have increased needs for self-care and more opportunities to use and practice the acquired skills independently. In addition to children with ID and ASD, difficulties in these routines are more often described in young children with cerebral palsy, since they need the help of an adult to complete the activities [29,36].

Children’s engagement in activities of daily life at home is correlated with the level of engagement in the preschool classroom and the routines in which they are expected to participate. Children with developmental disabilities did not completely follow the procedure of arriving at and departing from preschool, and they showed less respect for the routines of napping, listening to music, telling stories, and small group activities. They were less involved in outside circle activities, activities in the preschool block area, and in the performance of activities related to maintaining personal hygiene with the help of educators. 

Although children with developmental disabilities have the same needs to participate and enjoy various activities as their typically developing peers, the obtained results show that the level of their engagement, social relationships, and independence in the routines that take place in a preschool is lower, which is compatible with the findings of other research [37,38,39].

The tendency to engage in individual routines in preschool differs between groups and in regard to the structure of the routine itself. The biggest differences between the groups are observed in unstructured activities, which include play and interaction with peers outside and during free play. Free play provides opportunities for engagement, creativity, and interaction with others, which typically developed children have mastered to a great extent. The degree of organization of the game and the ability to understand and apply the rules of the game are of particular importance when a child with developmental disabilities is included in a peer game. The child’s engagement in these routines, in addition to developed communication and motor skills, requires skills such as joining in play with others, leading independent play, and being capable of functional play, which is often a challenge for these children. Therefore, children’s functionality may be related to their ability to follow these directions [26]. In terms of structured activities, there are significant differences between groups in activities that require a lower level of engagement, such as small group instruction activities and telling stories. In these activities, the teachers give various instructions, and the children follow the instructions describing what they must do, and the teachers are the ones who organize the activity. Typically developing children follow these rules and instructions more easily, unlike children with developmental disabilities, who have problems in understanding orders, remembering instructions, following directions, and realizing relationships. Therefore, children’s functionality may be related to their ability to follow these directions [26].

Children with developmental disabilities are most engaged in the practice of arriving at preschool and participating in meals. These are activities that have a clear structure that repeats itself in a predictable way, and thus the children feel safer. The level of children’s engagement in the activities of everyday life at home correlates with the activities in the preschool institution. Children with developmental disabilities face additional barriers that hinder their engagement in activities, such as a lack of initiative and support from adults to encourage the child’s engagement, a lack of instructional skills, insufficient use of opportunities, and negative societal attitudes [12,40]. At the same time, the relationship of typically developing children with children with disabilities in the same group is characterized by shorter and less frequent periods of communication, as well as interactions that are more frequent in directed activities [12]. 

Relationships and attitudes of this kind affect the formation of social relationships with peers and adults. Apart from activities related to meals and free play situations, children with developmental disabilities show a very low level of social relationships. Challenges in social relationships among children with disabilities are visible in situations involving a group of children and adults, where the children are expected to make contact, show initiative in solving problems, and participate in cooperative play. Embedding motivations of social interaction into the intervention, which can be developed through play in young children, may be a successful method for improving children’s social areas [41]. 

The functionality of children in both groups of respondents, observed through their independence in the activities of daily life at home and in the preschool classroom, has a very similar tendency, with an emphasis on much lower levels in children with developmental disabilities. The manifestation of the lowest level of children’s independence in activities related to music and meals was common to both groups, probably because these domains are the least represented in these routines. Both groups of children showed the highest levels of independence during outside play, since independence in this domain is expressed to the greatest extent through the child’s ability to move, climb, and run independently, which was not a dominant problem for the children included in this research, similar to previous research [42]. Children’s participation in various activities is influenced by the type of disability, and children with intellectual disability show less dependence on parental support compared to children with ASD and greater independence in social and leisure activities [43]. Young children with CP participate less often in family and recreational activities and require more help to participate in self-care than young children without CP [29]. In relation to the child’s current motor functions, in children with CP, the differences refer to the domain of self-care in everyday life but not to the family and recreational activities in the domain of a child’s engagement in everyday life [44].

A significant difference in independence between the two groups was observed in tooth-brushing behaviors. Although these are low-sophisticated activities, they require a high level of engagement and independence, probably due to the need to rely on the help of educators or parents. 

When comparing research result across countries, it should be kept in mind that there are differences between countries and, therefore, variations in the patterns of children’s engagement in different daily routines. These differences affect the interpretation of the child’s engagement in relation to the amount of time spent participating in an activity and with people or materials, as well as the environment in which the child is observed [38]. 

Limitations and directions for future research:

This study also has some limitations. The first limitation refers to the sampling process. That is, the phenomenon that not all the regions of the Republic of Serbia were included in the research, which should be ensured in subsequent research. Given that the results of some studies [13] indicate that an inclusive context, viewed not only through the quality of teacher–child interactions but also through the organization of spatial activities, contributes to child engagement, another limitation of this study is reflected in the fact that the subsamples of children were not separated in the data analysis based on the children who attended inclusive and development groups, i.e., the effects of the different characteristics of these environments (e.g., space, the quality of support that the child receives, relationships with adults, etc.) on the level of engagement were not verified. Future research should, therefore, also focus on understanding the relationship between environmental characteristics and engagement levels. Bearing in mind that this research did not verify whether there are differences in the engagement of children in relation to the type and level of disability, subsequent research should include a larger number of respondents and group them into sub-samples according to the mentioned variables. In addition, considering that this instrument has thus far been used for samples of children with cerebral palsy, there is the need for caution when interpreting the results of the raw scores obtained from the CEDL. At the same time, it represents an opportunity to examine its use among populations of children with other types of disabilities.

## 5. Conclusions

Despite the limitations, this study obtained results that provide significant information to practitioners regarding child and parental factors that contribute to children’s engagement in preschool and daily family activities and routines, as well as the areas in which the most pronounced differences in the engagement of children with developmental disabilities, compared to their typically developing peers, occur. Additionally, the value of this research is reflected in the verification and practical application of instruments for assessing children’s engagement, encouraging practitioners to use them in their daily work so as to monitor engagement, identify activities and routines in which the level of engagement is lower than expected, and plan support programs accordingly.

This study resulted from the project “Strengthening the ECI system in Serbia”, which is implemented by the Belgrade Psychological Center.

## Figures and Tables

**Figure 1 ijerph-19-14741-f001:**
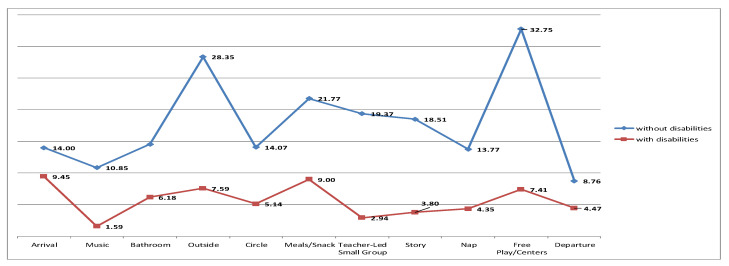
E = engagement.

**Figure 2 ijerph-19-14741-f002:**
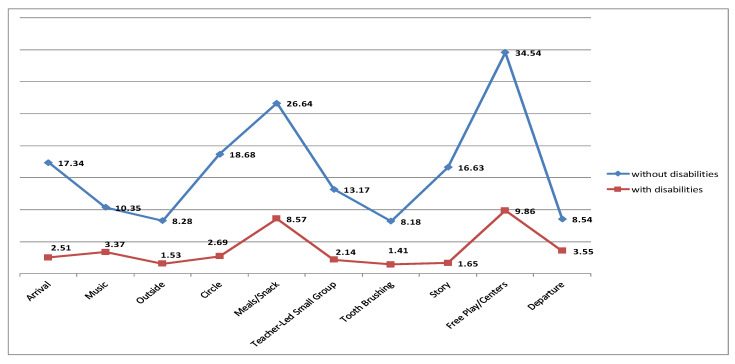
SR = social relationships.

**Figure 3 ijerph-19-14741-f003:**
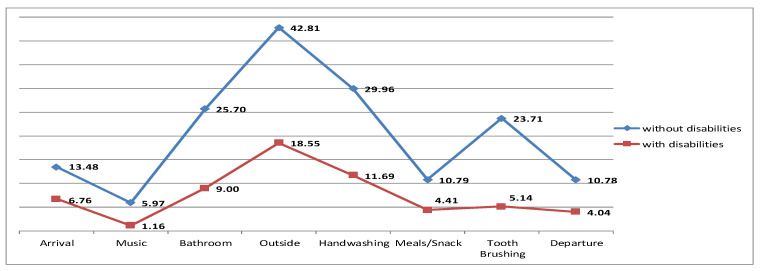
I = independence.

**Table 1 ijerph-19-14741-t001:** Sociodemographic characteristics of the study subjects.

Parents	
**Age** (years)	Range (24–64) 34.72 ± 6.02
**Gender**	
Male	40 (26.7%)
Female	110 (73.3%)
**Education level**	
Primary education (8 years in total)	9 (6.00%)
Secondary (11–12 years in total)	84 (56.00%)
Higher School and University (16–17 years in total + Master/PhD)	57 (38.00%)
**Employment status**	
Unemployed	113 (76.4%)
Employed	35 (23.6%)
**Marital status**	
Marriage	127 (84.67%)
Single	23 (15.33%)
Children	
**Age (years)**	Range (3–5) 4.02 ± 0.78
**Age categories**	
3 years	10 (5.3%)
4 years	79 (41.6%)
5 years	80 (42.1%)
**Gender**	
Male	92 (61.3%)
Female	58 (38.7%)
**Disability**	
No	49 (32.7%)
Yes	101 (67.3%)
**Number of siblings**	
	Range (0–5) 1.13 ± 0.84
**Number of household members**	
	Range (2–8) 4.05 ± 1.04

**Table 2 ijerph-19-14741-t002:** Descriptive Statistics and Alpha (α) Reliabilities for the Used Variables.

		Min	Max	Mean	SD	Sk	Ku	α	MIIC
**ClaMEISR**	Engagement total score	15.00	234.00	136.20	82.02	0.025	−1.81	0.950	0.774
	Social relationships total score	6.00	195.00	108.12	76.52	−0.039	−1.84	0.959	0.890
	Independence total score	6.00	192.00	118.67	63.68	−0.225	−1.53	0.930	0.802
**Child Engagement in Daily Life**	Frequency of Participation	28.10	100.00	65.89	15.53	0.496	−0.186	0.899	0.454
	Enjoyment of Participation	2.18	5.00	4.31	0.68	−1.30	0.818	0.897	0.451
	Participation in Self-Care	21.20	100.00	75.60	17.48	−0.643	0.422	0.880	0.524

Note: SK—skewness; Ku—kurtosis; α—Cronbach’s alpha coefficient of internal consistency reliability; mean inter-item correlation (MIC).

**Table 3 ijerph-19-14741-t003:** Correlations for the Analyzed Measures.

		1	2	3	4	5	6
**ClaMEISR**	Engagement total score	1					
	2. Social relationships total score	0.987 **	1				
	3. Independence total score	0.974 **	0.963 **	1			
**Child Engagement in Daily Life**	4. Frequency of Participation	0.659 **	0.629 **	0.652 **	1		
	5. Enjoyment of Participation	0.729 **	0.706 **	0.740 **	0.700 **	1	
	6. Participation in Self-Care	0.564 **	0.599 **	0.616 **	0.364 **	0.499 **	1
	Gender	0.254 *	0.224 *	0.192 *	0.079	0.133	0.169 *
	Age ^+^	0.328 **	0.355 **	0.433 **	−0.012	−0.095	0.310 **

Note: Pearson coefficients between pairs of measures; + partial correlation (adjusting for the existence of disabilities). **—*p* < 0.01, *—*p* < 0.05.

**Table 4 ijerph-19-14741-t004:** Differences in the ClaMEISR and CEDL subscales between children without disabilities and with disabilities.

Scale/Dimension		Without Disabilities		With Disabilities		t (df)	*p*	*D*
		Mean	SD	Mean	SD			
**ClaMEISR**	**Engagement** **total score**	**210.49**	**38.78**	61.92	28.72	21.55 (96)	0.000	3.27
	Social relationships total score	173.60	37.93	37.29	30.09	20.00 (100)	0.000	3.96
	Independence total score	169.34	33.59	60.76	32.77	16.72 (103)	0.000	4.35
**Child Engagement in Daily Life**	Frequency of Participation	72.11	14.72	54.23	8.91	8.96 ^a^ (137)	0.000	1.37
	Enjoyment of Participation	4.67	0.28	3.65	0.71	9.62 ^a^ (56)	0.000	2.13
	Participation in Self-Care	82.82	11.57	61.01	18.39	7.59 ^a^ (67)	0.000	1.54

ᵃ Levene’s test is significant (*p* < 0.05), suggesting a violation of the assumption of equal variances; Cohen’s d.

## Data Availability

The data presented in this study are available on request from the corresponding author. The data are not publicly available due to privacy.

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
