# Peer review of "Engagement of Preschool-Aged Children in Daily Routines"

_ijerph, 2022, doi:10.3390/ijerph192214741_

Round 1

Reviewer 1 Report

Overall, this paper is important to contribute cultural relevance to the testing of a measure within the Serbian context. The expectations of differences between children with and without disabilities and high internal consistency give sufficient data for the authors claims. They nicely summarize the study's limitations.

Minor typo on page 5 - "eacher-directed" rather than "teacher-directed." A bit more information on the functional assessment measure would be helpful; consider adding more information to the sentence "Items with > 3 were calculated." 

Author Response

Reviewer 1

Overall, this paper is important to contribute cultural relevance to the testing of a measure within the Serbian context. The expectations of differences between children with and without disabilities and high internal consistency give sufficient data for the authors claims. They nicely summarize the study's limitations. Minor typo on page 5 - "eacher-directed" rather than "teacher-directed." A bit more information on the functional assessment measure would be helpful; consider adding more information to the sentence "Items with > 3 were calculated." 

Authors:

First of all, we would like to thank the editor and the reviewer for some beneficial comments. All changes in the text are given in blue.

We fixed "eacher-directed"; we added t.

We have added information about the functional assessment.

We added: Data collection for functional assessment in early childhood can be realized through observation of the child (about the child's current functioning in activity contexts) and/or through a caregiver interview (via family input about child functioning).

We have added a more precise explanation about "Items with > 3 were calculated" in section 2.3. Instruments. Now it looks like this: “Different subscales have different numbers of items within functional domains of engagement, independence, social relationships. If a routine has < 3 items in a domain, the score of that routine is not calculated (for example, handwashing has only one engagement item and one social relationship item, so no scores should be calculated for those two domains in handwashing). 

Reviewer 2 Report

There are some spelling errors that need attention. Consideration should be given to the consent/ assent of the children when research was carried out with or without their understanding- See Alderson and Morrow: 

The ethics of research with children and young people a practical handbook

Alderson, Priscilla.Morrow, Virginia.

Los Angeles, Calif. ; London : SAGE2nd ed.2011  

Author Response

Reviewer 2

There are some spelling errors that need attention. Consideration should be given to the consent/ assent of the children when research was carried out with or without their understanding- See Alderson and Morrow: 

The ethics of research with children and young people a practical handbook

Alderson, Priscilla.; Morrow, Virginia. Los Angeles, Calif. ; London : SAGE; 2nd ed.; 2011  

Authors:

First of all, we would like to thank the editor and the reviewer for some handy comments. All changes in the text are given in blue.

We obtained parental consent for the study. We believe that it can satisfy the ethical criteria. We added: Considering that the research involves young children who do not have full insight into its structure and the contents, parents responsible for children’s care signed the informed consent on behalf of their children. Anonymity of research data has been ensured.

We are grateful for the referred literature

Reviewer 3 Report

Dear authors,

The article "Engagement of Preschool-Aged Children in Daily Routines", which aims to assess children's functioning in early childhood routines, is interesting and has a coherent structure. However, I would suggest some improvements:

The abstract clearly presents the aim of this research, the sample, the instruments used, as well as the final results and conclusions.

The keywords are logical and fit the object of the study.

The introduction has enough previous studies, some very current and others less so, to provide an understanding of inclusive education programmes developed in different contexts and specifically in schools in Serbia.

The sample is well described. 150 children with and without disabilities from schools in Serbia participated. Table 1 provides relevant information.

The procedure provides a favourable report from the Ethics Committee of the Faculty of Medicine. Participants were informed about the process at all times and were given the informed consent form.

The instrument has been accurately described, but is it validated? It is only mentioned that it has been translated into Serbian, but it would be useful to provide data on its validity and internal consistency.

The data analysis section is very well substantiated and detailed. Although it is indicated that the internal consistency and coherence of the instruments were calculated, it would be valuable to include these values briefly in the previous section, as this will be developed in detail in the results section. Or at least briefly indicate that these instruments have good psychometric properties which will be discussed later.

Results: high inter-item consistency and high values for the MIIC are indicated. Significant and strong correlations were found between the different variables proposed. The results revealed a statistically significant difference in the subscale ClaMEISR and CDLS depending on whether the child was developmentally disabled or non-disabled. Children without disabilities had higher scores in all domains, indicating better functioning. No differences on to the level of education were found in the group of parents, whereas differences on marital status and employment were found. The tables and graphs provided are clear and appropriate to the content shown.

In the discussion section the aim of this research is stated again and then the reflection on the results obtained is presented. The obtained results showed that both instruments applied have high internal consistency. Emphasis is placed on: Parents's role in enhancing their children's development is crucial from their birth with the exception of priorities regarding inclusion in daily routines of early childhood education, which vary according to child's age and stage of development. The results of this research confirm that children with developmental disabilities show a lower level of engagement and therefore participate less in family, recreational activities and self-care, but that they enjoy them just as much. Overall, this section is very well grounded, thoughtful and relates to the previous studies discussed in the introduction. There are some limitations which are discussed in a coherent manner.

In the references section, although they all conform to the MDPI standards, it is important to correct a generalised detail. The year of publication of articles is in bold and not in italics. Please correct all erroneous references.

Overall, the article presents interesting content for the scientific community. I do not doubt the value of this work.

Author Response

Authors:

First of all, we would like to thank the editor and the reviewer for some very useful comments. All changes in the text are given in blue.

According to the reviewer's suggestion, we added one sentence about psychometric characteristics at the end of section 2.3. Instruments: “The present study showed good psychometric properties of these instruments, which will be further discussed.

We have corrected all errors in the references.